# TRAINING EQUILIBRIA IN REINFORCEMENT LEARNING

## ABSTRACT

In partially observable environments, reinforcement learning algorithms such as policy gradient and Q-learning may have multiple equilibria—policies that are stable under further training—and can converge to equilibria that are strictly suboptimal. Prior work blames insufficient exploration, but suboptimal equilibria can arise despite full exploration and other favorable circumstances like a flexible policy parametrization. We show theoretically that the core problem is that in partially observed environments, an agent's past actions induce a distribution on hidden states. Equipping the policy with memory helps it model the hidden state and leads to convergence to a higher reward equilibrium, *even when there exists a memoryless optimal policy*. Experiments show that policies with insufficient memory tend to learn to use the environment as auxiliary memory, and parameter noise helps policies escape suboptimal equilibria.

## 1 INTRODUCTION

In *Markov decision processes* (MDPs), Q-learning and policy gradient methods are known to always converge to an optimal policy (Watkins and Dayan 1992; Bhandari and Russo 2019) subject to the assumptions detailed in Section 3. In non-Markovian environments such as *partially observable MDPs* (POMDPs), this guarantee fails when using Markovian policies: the algorithms don't always converge to an optimal policy, and there may be multiple (suboptimal) policies surrounded by 'basins of attraction'. This is true even in the tabular setting with full exploration, where formal convergence guarantees are strongest.

For example, consider the *double-tap* environment described in Figure 1. The optimal policy in this environment is to always choose action $a_2$. However, a policy that favors $a_1$ is stuck in a local optimum. Thus this policy will never converge to the optimal policy even with further training, despite exploring the full state-action space.

We call a policy that is fixed under further training an *equilibrium* of the training process. For policy gradient methods, equilibria correspond to stationary points of the expected return; for Q-learning, equilibria are fixed points of the Bellman equation. Pendrith and McGarity (1998) construct an environment in which Q-learning may converge to a suboptimal policy. We extend their finding and show that in POMDPs, policy gradient methods and Q-learning may have multiple training equilibria and sometimes converge to a suboptimal equilibrium despite full exploration and other favorable circumstances (described in Section 3).

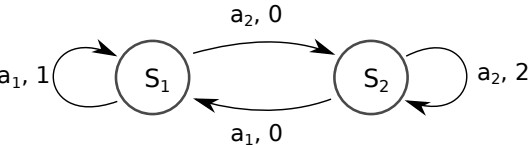

Figure 1: The *double-tap* environment. Arrows are transitions between states, labeled with a corresponding action and reward. The agent is rewarded when it chooses the same action twice in a row. The states $s_1, s_2$ are unobserved. The optimal policy is to always choose action $a_2$. However, a policy that favors $a_1$ is caught in a local optimum, since picking $a_2$ is only profitable once done with high probability (see Figure 2a).

**Theoretical results.** We show that multiple equilibria can only emerge when the distribution of unobserved states depends on past actions (Proposition 3.1). Our interpretation is that a memoryless policy must play a coordination game with its past self, and may get trapped in suboptimal Nash equilibria. For some contexts, we formalize this interpretation and show that Nash equilibria of coordination games correspond to the training equilibria of an associated RL environment (Theorem 3.2). A sufficient amount of memory resolves the coordination game and provably implies a unique optimal training equilibrium (Proposition 4.1). Thus memory can be crucial for *convergence* of RL algorithms even when a task does not require memory for optimal performance.

**Empirical results.** (Reported in Section 5). We confirm empirically that our theoretical results hold in practice. In addition, we show that even a memoryless policy can often learn to use the external environment as auxiliary memory, thus improving convergence in the same way that policy memory does. However, there exist counterexamples in which even a flexible environment that allows for external memory is not sufficient to learn an optimal policy. We also confirm a hypothesis that in environments with multiple equilibria, parameter noise (Rückstieß et al. 2008; Plappert et al. 2018) can lead to convergence to better equilibria, thus providing a novel explanation of why parameter noise is observed to be beneficial.

## 2 BACKGROUND ON POMDPS

Reinforcement learning (RL) (Sutton and Barto 2018) is the problem of training an agent that takes actions in an environment in order to maximize a reward function. The most common formalism for reinforcement learning environments is the *Markov decision process* (MDP). An MDP models an environment which is Markovian in the sense that the environment state $s_t$ and reward $r_t$ at time $t$ depend only on the previous state $s_{t-1}$ and the previous action $a_{t-1}$[1]. Crucially, in this formalism a policy has access to the entire state $s_t$ at each step, and thus no memory is necessary to perform optimally.

Most realistic environments are not MDPs, since real-world problems are invariably *partially observable*. For example, a driving agent must anticipate the possibility of other drivers emerging from side-roads even if they are currently out of sight. To model partial observability, it is common to extend the MDP formalism to define a *partially observable Markov decision process* (POMDP) (Åström 1965). The main idea is to model the environment as an unobserved MDP, and allow the policy to access observations sampled from an observation distribution $O(o \mid s)$ conditional on the current state.

Formally, a POMDP is a tuple $(\mathcal{S}, \mathcal{A}, \mathcal{O}, T, O, R, \gamma, \eta_0)$, where $\mathcal{S}$ is the set of states, $\mathcal{A}$ the set of possible actions, $\mathcal{O}$ the set of observations, $T$ the transition kernel, $O$ is the conditional distribution of observations, $R$ is the reward function, $\gamma \in [0, 1)$ is the discount factor, and $\eta_0$ is the initial state distribution. Let $s_t$ denote the state at time $t$. Then a timestep proceeds as follows: an observation is drawn according to the distribution $O(o_t \mid s_t)$ and given as input to the policy, the policy outputs an action $a_t$, the agent receives reward $R(s_t, a_t)$, and the next state is generated according to the transition kernel $T(s_{t+1} \mid s_t, a_t)$.

## 3 TRAINING EQUILIBRIA IN POMDPS

It is well-known that Q-learning and policy gradient methods converge to a globally optimal policy if the environment is a (fully observable) MDP (Watkins and Dayan 1992; Bhandari and Russo 2019). However, in partially observable environments we have no such general guarantees. In particular, in this section we will study partially observable environments where RL algorithms *do* converge, but there exist multiple policies—training equilibria—to which they might converge, some of which are suboptimal.

It is also possible that a training algorithm may not converge at all. Q-learning with discontinuous action-selection methods[2] may end up oscillating between two suboptimal policies (Gordon 1996). However, it is known that for Q-learning with continuous action selection, fixed points always exist

---

[1] In some formulations, the reward may also depend on the current state $s_t$.

[2] For example $\varepsilon$-greedy action selection (Sutton and Barto 2018).

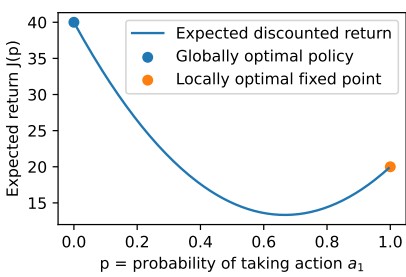 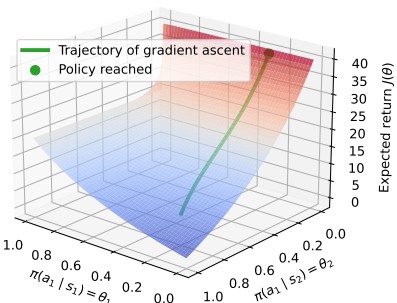

(a) Expected return of memoryless policy. Gradient ascent on expected return $J(p)$ converges to one of the extremes $p = 0$ or $p = 1$. A low probability of $a_1$ is preferable, but an agent that favors $a_1$ can stay stuck in the local maximum (orange).

(b) Expected return of a policy with a one-step memory $m^{(t)} = a^{(t-1)}$. The return is a function of policy parameters $\theta_1$ and $\theta_2$. This function has an easily reachable global optimum at $\theta_2 = \pi(a_1 \mid s_2) = 0$ and $\theta_1 < 1$, and no suboptimal fixed points.

Figure 2: The expected return of parametrized policies in the double-tap environment from Figure 1. An optimal solution to this task requires no memory, yet the optimal policy is more easily reached by a policy with memory. In both cases we use discount factor $\gamma = 0.95$, though the results are not sensitive to the value of the discount factor.

(Perkins and Pendrith 2002). In summary, it is possible for training algorithms to not converge at all, but this is a relatively rare circumstance that can be mitigated. In the rest of this work, we will assume we are working in a setting in which convergence is guaranteed, though not necessarily to an optimal policy.

To avoid 'spurious' convergence problems, we will introduce the following assumptions on the environment and the learning algorithm. In the fully observable setting, theses assumptions are enough to guarantee convergence of Q-learning and policy gradient methods to an optimal policy. As we will see, in partially observed environments this is not the case.

**Assumption 3.1** (Full exploration). *Every state-action pair is seen infinitely often during training.*

**Assumption 3.2** (Flexible policy parametrization). *The policy space is closed under policy improvement.*[3]

**Assumption 3.3** (Bounded rewards). *There exists some $R_{max}$ such that for all state-action pairs $(s, a) \in \mathcal{S} \times \mathcal{A}$ we have $R(s, a) \leq R_{max}$.*

**Assumption 3.4** (Robbins-Monro). *For all state-action pairs $(s, a)$, the learning rate $(\alpha_n)_{n \geq 0}$ satisfies $\sum_{i=0}^{\infty} \alpha_{n(i,s,a)} = \infty$ and $\sum_{i=0}^{\infty} \alpha_{n(i,s,a)}^2 < \infty$. Here, $n(i, s, a)$ is the index corresponding to the $i$th time that the agent visits state $s$ and takes action $a$.*

### 3.1 TRAINING EQUILIBRIA

We can view a training algorithm as a stochastic mapping $A$ that maps a policy $\pi_k$ onto an improved policy $\pi_{k+1} = A(\pi_k)$.

*Example* 3.1 (Idealized policy gradient). Idealized vanilla policy gradient is a training algorithm on the space of parametrized policies $\Pi = \{\pi_\theta; \theta \in \Theta\}$ that maps a policy $\pi_{\theta_k}$ to $A(\pi_{\theta_k}) = \pi_{\theta_{k+1}}$, where

$$\theta_{k+1} = \theta_k + \eta \nabla R(\theta_k),$$

$R$ is the expected return, and $\eta > 0$ is a step-size hyperparameter.

Any training algorithm induces a stochastic process in the policy-space $\Pi$ given by $(\pi_0, \pi_1, \dots)$. It is natural to ask what the fixed points of this process are. A *training equilibrium* of a training algorithm

---

[3]Tabular policies satisfy this requirement, as do neural networks if they are flexible enough to represent any tabular policy.

$A$ is then a policy $\pi \in \Pi$ that is in expectation held fixed by the training algorithm: $\mathrm{E}[A(\pi)] = \pi$. For example, the training equilibria of idealized vanilla policy gradient are all the policies $\pi_\theta$ such that their parametrization satisfies $\nabla R(\theta) = 0$, that is the class of stationary points of the expected return in parameter space.

In the following we will be study two training algorithms in particular: REINFORCE, also called 'vanilla policy gradient' (Williams 1992; Achiam 2018) and Q-learning (Watkins and Dayan 1992). We consider the setting where all Assumptions 3.1 through 3.4 are satisfied. Under these conditions it follows from standard convergence results (Bhandari and Russo 2019; Watkins and Dayan 1992) that in MDPs the set of training equilibria is exactly the set of optimal policies. This is *not* the case in POMDPs, as we show via counterexample in Section 3.2.

The core problem is that the conditional distribution $\eta^\pi(s \mid o)$ of the hidden state given an observation can vary depending on past actions, thus $\eta^\pi$ depends on the policy. Indeed, when $\eta$ does *not* depend on past actions, then both policy gradient and Q-learning algorithms are guaranteed to converge to an optimal policy.

**Proposition 3.1** (Optimality in POMDPs, informal). *Assume that the conditional distribution $\eta^\pi(s|o)$ of the POMDP state given an observation is independent of the policy in the sense that there exists a distribution $\eta$ such that for all state-observation pairs $(s, o)$ and all policies $\pi$, $\eta(s \mid o) = \eta^\pi(s \mid o)$. Then if conditions 3.1-3.4 hold, Q-learning and policy gradient methods converge to an optimal policy.*

A rigorous statement and proof of Proposition 3.1 is given in Appendix C.1. Having identified dependence on past actions as the source of convergence problems, we now discuss a simple example of an environment a suboptimal training equilibrium.

### 3.2 THE 'DOUBLE-TAP' ENVIRONMENT

This environment is introduced in Figure 1. The optimal policy is to always choose action $a_2$. However, a policy that favors $a_1$ is stuck in a local optimum, since picking $a_2$ is only profitable once done with high enough probability (see Figure 2a).

Indeed, consider applying a policy gradient method (Sutton, McAllester, et al. 2000) to the double-tap environment. Policy gradient algorithms use stochastic gradient descent to maximize the discounted return $J(\theta) = \mathrm{E}\left[\sum_{t=0}^\infty \gamma^t r_t\right]$, where $r_t$ is the reward received in timestep $t$.[4] Since there are no observations in this environment, we can define a policy $\pi$ using only a single parameter $p$, the probability of taking action $a_1$:

$$\pi(a) = \begin{cases} p & \text{if } a = a_1 \\ 1 - p & \text{otherwise.} \end{cases}$$

The expected single-step return for this policy is[5]

$$\mathrm{E}_\pi[r] = \Pr(s = s_1 \text{ and } a = a_1) + 2\Pr(s = s_2 \text{ and } a = a_2)$$
$$= p^2 + 2(1 - p)^2.$$

The full discounted reward is then given by $J(p) = (p^2 + 2(1 - p)^2)/(1 - \gamma)$, plotted in Figure 2a. As is visible in this plot, gradient ascent may get stuck in the local maximum at $p^* = 1$ if initialized poorly at some $p_0 > 2/3$.

In terms of the environment dynamics, this is because actions not only determine the reward in the current timestep but also affect future state: choosing action $a_1$ always leads to state $s_1$. The policy does not know which state it is in, and so needs to choose an action that works well on average. If the environment is in state $s_1$ with high probability then it is best to pick action $a_1$, which reinforces the suboptimal equilibrium.

---

[4]In non-episodic problems the objective function of policy optimization methods is the *average rate of reward* $\lim_{T \to \infty} 1/T \sum_{t=0}^T \mathrm{E}[r_t]$ rather than the full discounted return (Sutton and Barto 2018, chapter 13.6). This amounts to the same thing in our case, since the rate of reward is constant after the first timestep.

[5]This isn't quite true for the first reward, where the value is simply $p$ or $2(1 - p)$, depending on whether we start in state $s_1$ or $s_2$. We ignore this for simplicity, since the effect on the training dynamics is negligible.

A straightforward solution is then to augment the policy with memory: if the policy can condition on the action it took in the previous timestep, it will take action $a_2$ conditional on having done the same in the previous timestep, thus making $a_2$ a better option than $a_1$. We discuss this approach in Section 4.

### 3.3 CORRESPONDENCE TO NASH EQUILIBRIA AND SELF-PLAY

A Nash equilibrium (Nash Jr 1950) is a solution to a non-cooperative game, in which no player can increase their payoff by switching to a different strategy so long as the other player's strategies are held fixed.[6] A crucial assumption in such games is that decisions are made simultaneously: no player has knowledge of the other player's move before they move. If this assumption is broken, then the player that goes first could trigger a departure from the Nash equilibrium by playing a non-equilibrium strategy.

This is reminiscent of the POMDP setting, in which a policy has no knowledge of its own past (or future) history and thus is stuck in an equilibrium determined by its own past actions. Just like in the game-theoretic setting, when a policy is allowed to condition on its past it no longer gets stuck in suboptimal equilibria (Section 4).

In the following we make this analogy formal in two ways. First we show that for every two-player game, there exists a corresponding RL environment with the same dynamics, and that the training equilibria of the environment correspond to the Nash equilibria of the game. Second, we point to results from the literature on self-play in reinforcement learning, which shows that RL training converges to a Nash equilibrium.

**Two-player games.** Any two-player game can be turned into a 'single-player' RL environment by having the RL agent play against a fixed policy. In particular we can choose a 'best-response' opponent policy that is able to see the history (in contrast to the trained policy, which is memoryless), and always chooses the action that would have performed best against previous actions within an episode. An example of this kind of environment is the *Battle of the Sexes* environment introduced in Section 5. A trained policy will converge to a Nash equilibrium, and there is a one-to-one correspondence between Nash equilibria and training equilibria.

**Theorem 3.2** (Informal). *Let $E$ be a POMDP constructed from a two-player game $G$ by letting the policy play against a best-response strategy. Then the following are equivalent:*

  *(i) The policy $\pi$ is a policy-gradient fixed point in $E$.*

  *(ii) The policy $\pi$ is part of a Nash equilibrium in $G$.*

A formal statement and proof of Theorem 3.2 is available in Appendix B.

**Self-play.** In general, equilibria can arise when an agent interacts with other agents that are also learning, or copies of itself (i.e. agents that share the same policy). Indeed it is well known that multi-agent RL and self-play converge to a Nash equilibrium that is not necessarily Pareto-optimal (Conitzer and Sandholm 2003).

We have shown two formal results that highlight the correspondence between game-theoretic Nash equilibria and RL training equilibria. In practice, the analogy seems to extend to contexts outside of self-play or those covered in Theorem 3.2. Nash-equilibrium-like dynamics can arise whenever an agent's past actions affect the environment in a way that the agent cannot condition its future actions on. When this happens, the agent effectively interacts with its past self and typically ends up in a Nash equilibrium of a cooperative game. For example, the double-tap environment is equivalent to a coordination game **??** in which two copies (agents that share the same policy) each take an action at every timestep: both get reward 1 if they both take action $a_1$ and reward 2 if they both take action $a_2$, and reward 0 otherwise. (We invite the reader to check that this indeed leads to the same dynamics).

---

[6]We provide a complete definition in Appendix B.

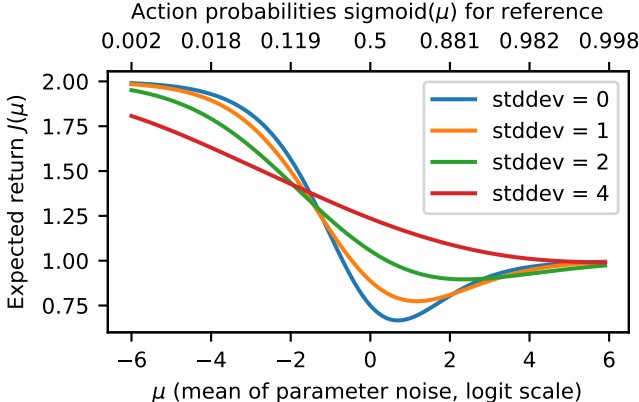

Figure 3: The effect of parameter noise on the curve of expected return in the 'double-tap' environment. The blue curve (no parameter noise) is identical to the curve in Figure 2a transformed to logit scale. With increasing noise scale the bad local maximum is 'smoothed out' and gradient ascent can more easily converge, even when initialized very unfavorably.

## 4 MITIGATING FACTORS: MEMORY AND PARAMETER NOISE

### 4.1 MEMORY CAN ELIMINATE SUBOPTIMAL LOCAL OPTIMA

In MDPs, it is standard to work with policies that are *memoryless*: the action $a_t$ depends only on the current observation $o_t$, not on any previous observations or actions. Many partially observable environments however require memory for optimal play. A policy with memory samples an action from a conditional distribution $\pi(a_t \mid o_t, m_t)$, where $m_t$ is the memory state at time $t$. In the limit, the memory might contain the entire history: $m_t = (o_0, a_0, o_1, \ldots, a_{t-1})$. In this case we say that the policy has *unbounded memory*. In deep RL practice, it is common to use recurrent policies that keep track of a hidden state which encodes relevant past observations, or to concatenate several observations into a single input to the policy.

We find that memory is useful even when an optimal policy does not require memory to solve a task. This is because policies with (unbounded) memory confer the same convergence guarantees to RL algorithms in POMDPs that hold in the MDP setting.

**Proposition 4.1** (Policy memory implies convergence to an optimal equilibrium (informal)). *Assume the conditions 3.1-3.4 hold. If a policy has unbounded memory, then Q-learning and policy gradient methods are guaranteed to converge to an optimal policy.*

*Remark* 4.1. If only the past $k$ timesteps are relevant to the environment dynamics and the reward, then a policy memory of maximum length $k$ is enough to obtain the same guarantee.

Formal statements and proofs of Proposition 4.1 and Remark 4.1 are available in Appendix C.2.

In the double-tap environment, switching to a policy with memory (one timestep is enough) removes suboptimal local maxima from the graph of the expected return. This is shown in Figure 2b, where we use a two-parameter policy with a one-step memory. Details on the parametrization and the calculation of the expected return are available in Appendix A.1.2.

**Why not use memory?**   A practical reason not to use memory is that memoryless policies are simpler to implement, and less expensive to train and perform inference on. Moreover, the principle of least privilege (Saltzer and Schroeder 1975) recommends that systems should only be given the minimal capabilities needed. So, if a memoryless policy can achieve the task at hand, it should be favored over a policy with memory. For example, memoryless policies cannot leak information from a previous interaction with another user. Memoryless policies are also easier to test and validate, as they depend only on the current state, rather than the entire history of states – an exponentially larger space.

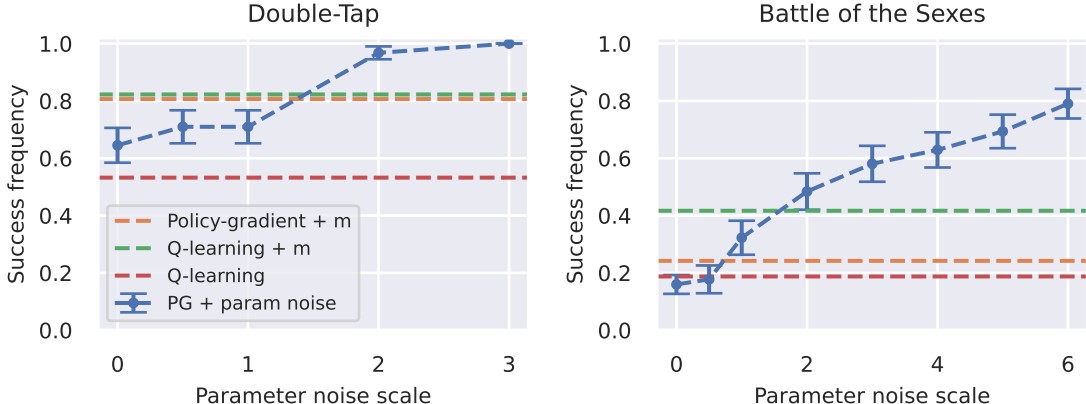

Figure 4: We confirm that parameter noise is beneficial in the Double-Tap (left) and the Battle of the Sexes environment (right). Blue: parameter noise is beneficial for tabular policy gradient converge. Green: Q-learning with external memory does well. Red: a memoryless policy does badly.

## 4.2 PARAMETER NOISE CAN ELIMINATE SUBOPTIMAL LOCAL OPTIMA

Since it may not always be feasible or desirable to equip policies with sufficient memory for convergence, we investigate Unfortunately, it is not feasible to equip policies with unbounded memory, we investigate parameter noise for policy gradient methods, which can alleviate the convergence failures outlined in Section 3. Since training equilibria correspond to stationary points of the expected return, it is plausible that more exploration in *parameter space* is helpful. This is in contrast to exploring in the *action space*, which Q-learning and VPG already do via stochastic policies. In other words, parameter noise explores at the level of strategies ('always take action $a_2$') rather than individual actions ('take action $a_2$ in timestep $t$').

*Parameter noise* (Rückstieß et al. 2008; Plappert et al. 2018; Fortunato et al. 2018) samples a parameter setting $\theta \sim \mathcal{N}(\mu, \Sigma)$ at the beginning of an episode, then hold that parameter fixed for the duration of the episode. At the end of an episode, we update the mean $\mu$ via gradient descent:

$$\nabla_\mu \mathrm{E}_\theta \left[ J(\theta) \right] = \mathrm{E}_\varepsilon \left[ \nabla_\mu J(\mu + \varepsilon^T \Sigma^{1/2}) \right] = \mathrm{E}_\varepsilon \left[ (\nabla J)(\mu + \varepsilon^T \Sigma^{1/2}) \right], \qquad (1)$$

where $\varepsilon \sim N(0, I)$ and we use the reparametrization trick (Kingma and Welling 2014) in the first equality. The covariance $\Sigma$ is held fixed.

As is visible in Figure 3, adding parameter noise effectively 'smoothes out' the curve of the expected return, making an optimal policy easily reachable even for a memoryless policy. We confirm this empirically in Section 5.

## 5 EXPERIMENTS

### 5.1 DOUBLE-TAP

Our first experiment is a sanity check: we confirm experimentally that the properties we show analytically for the double-tap environment in Section 3 also hold in practice. Our results in Figure 4 confirm that memoryless policies (Q-learning, in red, and policy gradients, in blue at $x = 0$) often fail to converge to the optimal policy. Parameter noise is helpful for policy gradients, with success frequency increasing with the standard deviation of the noise. In addition, we study augmenting the environment with *external memory*: we augment the action, state, and observation space by allowing the agent to flip a bit in addition to its other actions. Every timestep, it observes the value of the bit. The policy itself is still memoryless, but it can learn to use this external memory to perform better on both the environments we study. Results are shown in Figure 4.

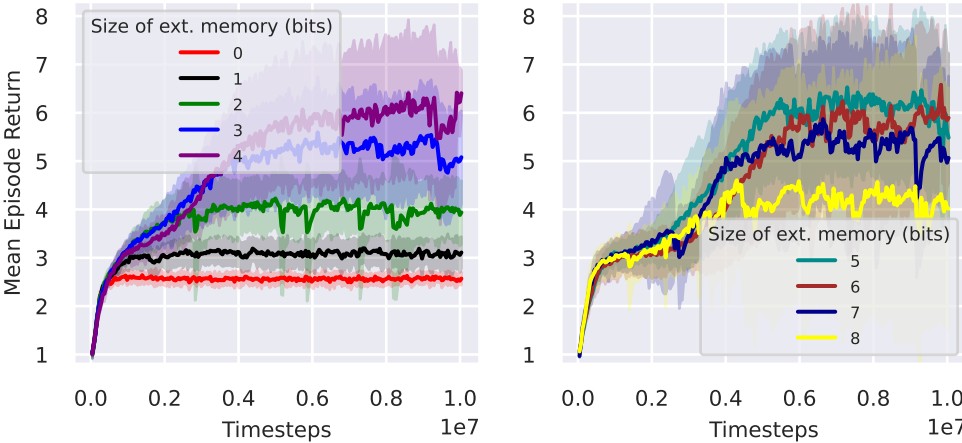

Figure 5: In the sequential bandit environment, PPO benefits from external memory. Left: more external memory brings additional benefit. Right: after a certain point, the complexity cost of additional external memory outweighs the marginal benefit.

## 5.2 BATTLE OF THE SEXES

*Battle of the Sexes* is a game in which two players must coordinate—without communication—on which concert to go to: Bach or Stravinsky. The two players have opposed preferences, but both prefer being together to being alone. This results in a payoff matrix described in Table 1. Note that this game has two Nash equilibria: (Bach, Bach) and (Stravinsky, Stravinsky).

We turn this game into an RL environment by letting the agent choose the actions of Player 1 and have Player 2 be controlled by the environment. Concretely, the policy action space is $\mathcal{A} = \{\text{Bach}, \text{Stravinsky}\}$, and the reward is equal to Player 1's payoff in Table 1. The actions of Player 2 are chosen by the environment to achieve maximal return for Player 2 on average given the agent's past actions in the episode: at timestep $t > 0$, Player 2 chooses action

$$a_t^{\text{Player 2}} = \text{argmax}_{a \in \mathcal{A}} \sum_{t' < t} r^{\text{Player 2}}(a_{t'}, a),$$

where $r^{\text{Player 2}}(a', a)$ is the payoff for Player 2 given that Player 1 takes action $a'$ and Player 2 takes action $a$. Player 2's first action is chosen uniformly at random.

The two Nash equilibria of the original game correspond to training equilibria of Q-learning or policy gradient. As discussed in Section 3.3, it is possible to generalize this notion to show that for every two-player game, Nash equilibria correspond to stationary points of the expected return.

|  |  | Player 2 | |
|---|---|---|---|
|  |  | Bach | Stravinsky |
| Player 1 | Bach | 1, 2 | 0, 0 |
|  | Stravinsky | 0, 0 | 2, 1 |

Table 1: Payoff matrix for Battle of the Sexes. Entries are tuples $r_1, r_2$, where $r_1$ and $r_2$ are payoffs for player 1 and player 2 respectively.

## 5.3 SEQUENTIAL BANDIT PROBLEM

We now study how well external memory works in practice. In all our prior environments there exists an optimal policy that is memoryless. We now introduce a new environment in which this is not the case: the *sequential bandit* environment. In this environment there are ten actions $\mathcal{A} = \{a_1, \ldots, a_{10}\}$, and an episode is ten steps long. The environment returns reward 1 for every new action taken *in*

*order* starting from $a_1$, that is for all $k \in \{1, \ldots, 10\}$,

$$R(s, a_k) = \begin{cases} 1 \text{ if actions } a_1, \ldots, a_{k-1} \text{ have already been taken in this episode} \\ 0 \text{ otherwise.} \end{cases}$$

To reliably execute the optimal policy (in timestep $t$, take action $a^{(t)} = a_t$), the agent needs to remember which actions it has already taken.

We study how a memoryless policy fares in an augmented version of this environment. As in Section 5.1, we augment the environment with external memory by allowing the agent to individually flip $n$ bits in addition to its other actions. Every timestep, it observes the values of these bits. For values of $n = 4$ or larger, an optimal policy ought to be in principle capable of tracking all ten actions and achieving perfect performance.

Results from training a PPO (Schulman et al. 2017) agent with a feedforward policy are shown in Figure 5. The policy reliably learns to make use of the external memory. However, it does not use it optimally: even an agent given a 4-bit external memory does not achieve the optimal reward of 10. For low values of $n$, a larger external memory is reliably better, while for large values of $n$ a larger external memory may degrade performance. We hypothesize that this is due to the added complexity as the size of the state and action space grows exponentially in $n$.

## 6    RELATED WORK

**Non-convergence in POMDPs.**   It is well-known that finding optimal policies in POMDPs is usually harder than in MDPs. In particular, Singh et al. (1994) emphasize the necessity of stochastic policies in POMDPs, and Littman (1994) show that in general, finding an optimal policy in a POMDP is NP-complete. Pendrith and McGarity (1998) discuss the existence of suboptimal training equilibria in Q-learning. Their analysis does not cover policy gradient methods, nor the benefits of parameter noise or external memory. Similarly, Chades et al. (2002) discuss training equilibria that can arise in multi-agent systems, which are necessarily partially observable. In the context of constrained policy classes, Bhandari and Russo (2019) discuss the failure of policy gradient methods to converge in MDPs and use an environment similar to our double-tap environment as example.

**Memory.**   Peshkin et al. (1999), Littman (1993), and Jaakkola et al. (1994) propose augmenting an environment with external memory in the context of POMDPs, though none of them discuss suboptimal local optima. More recently, Icarte et al. (2020) propose to make learning external memory more efficient by providing an action that pushes the current observation onto a "memory" stack. They also show by counterexample that policies do not always learn to effectively use external memory.

**Parameter noise.**   Most exploration methods in RL explore the state-action space. Parameter noise (Rückstieß et al. 2008) allows for additionally exploring the parameter space. Plappert et al. (2018) and Fortunato et al. (2018) show that parameter noise can be beneficial in high-dimensional continuous-control environments. We provide a novel potential explanation for these observations: in partially observed environments, parameter noise can help escape suboptimal training equilibria.

## 7    CONCLUSION

It is well-known that RL convergence guarantees require adequate exploration. In this work, we have studied a distinct but common failure mode: multiple training equilibria for Q-learning and policy gradient in POMDPs. We show that this problem is solved by unbounded memory, and may be alleviated by bounded memory (whether internal to the policy, or external in the environment). We also show the problem may be resolved by parameter noise, providing a novel explanation of the success of parameter noise in improving convergence of RL training.

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

# A  ENVIRONMENT DESCRIPTIONS

## A.1  DOUBLE-TAP

### Q-LEARNING ALSO HAS MULTIPLE FIXED POINTS

This section shows that just like policy gradient methods, Q-learning also may get stuck in the same suboptimal local optimum.

In contrast to policy optimization, Q-learning methods learn to approximate the action value function $Q^\pi$ and act $\varepsilon$-greedily based on this estimate. In particular we will cover *temporal difference Q-learning*. This algorithm samples a transition $(o, a, r, o')$ based on the current policy and then updates the action-value $q(o, a)$ via

$$q(o, a) = q(o, a) + \eta(r + \gamma \max_{a'} q(o', a') - q(o, a)) \tag{2}$$

We call $q^*$ a fixed point if the update in Equation 2 is equal to zero in expectation. Formally this means that for all observations and actions $o, a$,

$$q^*(o, a) = \mathop{\mathrm{E}}_{s, o' \sim \pi_{q^*}} [r(s, a) + \gamma \max_{a'} q^*(o', a')], \tag{3}$$

where the expectation is taken with respect to $s$ and $o'$ sampled from the distribution induced (in the limit of $t \to \infty$) by the policy $\pi_{q^*}$ that acts $\varepsilon$-greedily with respect to the parameters $q^*$. Such a fixed point might not be stable: since the update (2) is stochastic, a random change in parameters could induce a switch to a different policy. This sort of switch becomes less likely the as the difference between the value $\max_{a'} q(o, a')$ of the optimal action and the value of the next-best action grows.

We will now show that for the environment described by Figure (TODO ref figure) there are multiple mappings $q(o, a)$ that satisfy the fixed point Equation 2 Since in this example there is only a single, fixed, observation, we omit it from our notation and write $q(a)$ for the action-value. We assume that we are using $\varepsilon$-greedy exploration, so there are only two possible policies (assuming that $q(a_1) = q(a_2)$ rarely enough that we can safely ignore it):

- the policy $\pi_1$ which chooses $a_1$ with probability $1 - \varepsilon/2$ and $a_2$ otherwise.
- the policy $\pi_2$ which chooses $a_2$ with probability $1 - \varepsilon/2$ and $a_1$ otherwise.

It will be useful to write $s \sim \pi_i$ for the distribution of states that arises when acting according to policy $\pi_i$. We then write $r_i = \mathrm{E}_{s \sim \pi_i}[r(s, a_i)]$ for the expected single-step reward for choosing action $a_i$ if the agent has been acting according to the policy $\pi_i$ in the past. The rewards specified in Figure are $r(s_1, a_1) = 1$ and $r(s_2, a_2) = 2$ (all other rewards are zero), so this amounts to

$$r_1 = (1 - \varepsilon/2) \cdot 1, \text{ and}$$
$$r_2 = (1 - \varepsilon/2) \cdot 2.$$

**Proposition A.1.** *Assume that the exploration parameter satisfies $\varepsilon < 2/3$. Then the action-values*

$$q_1(a) = \begin{cases} r_1/(1 - \gamma) & \text{if } a = a_1 \\ \varepsilon + \gamma r_1/(1 - \gamma) & \text{otherwise,} \end{cases}$$

*and*

$$q_2(a) = \begin{cases} \varepsilon/2 + \gamma r_2/(1 - \gamma) & \text{if } a = a_1 \\ r_2/(1 - \gamma) & \text{otherwise,} \end{cases}$$

*both satisfy the fixed-point equation (3). Furthermore, $q_1$ induces policy $\pi_1$ while $q_2$ induces $\pi_2$.*

*Proof.* We will begin by showing that $q_1(a_1) > q_1(a_2)$ (and the converse for $q_2$). First, note that

$$q_1(a_2) = \varepsilon + \gamma q_1(a_1).$$

Thus $q_1(a_2) < q_1(a_1)$ if and only if

$$(1 - \gamma)q(a_1) > \varepsilon.$$

Now, $(1 - \gamma)q(a_1) = 1 - \varepsilon/2$, so this is equivalent to the condition

$$1 - \varepsilon/2 > \varepsilon$$
$$\iff 2/3 > \varepsilon,$$

which is satisfied by assumption. A similar computation shows that $q_2(a_2) > q_2(a_1)$.

We now show that $q_1$ is indeed a fixed point, meaning that for all actions $a$,

$$q_1(a) = \mathop{E}_{s \sim \pi_1} [r(s, a) + \gamma \max_{a'} q_1(a')].$$

A simple computation is enough: for $a_1$ we have

$$\mathop{E}_{s \sim \pi_1} [r(s, a_1) + \gamma \max_{a'} q_1(a')] = r_1 + \gamma q(a_1)$$
$$= r_1 + \gamma r_1/(1 - \gamma)$$
$$= r_1/(1 - \gamma)$$
$$= q_1(a_1).$$

Under $\pi_1$, the probability of being in state $s_2$ is $\varepsilon/2$. Since $r(s_2, a_2) = 2$, the expected single-step reward $E_{s \sim \pi_1}[r(s, a_2)]$ of taking action $a_2$ is exactly $\varepsilon$. Thus

$$\mathop{E}_{s \sim \pi_1} [r(s, a_2) + \gamma \max_{a'} q_1(a')] = \varepsilon + \gamma q_1(a_1) = q_1(a_2)$$

as desired. By an an analogous argument, $q_2$ is a fixed point as well. □

*Remark* A.1. If the exploration parameter $\varepsilon$ is close to $2/3$, then the action-values $q(a_1)$ and $q(a_2)$ are also close. In this case the suboptimal fixed-point may be unstable in the sense that the noise in the updates can 'dislodge' the policy.

### A.1.1 EXPECTED RETURN OF A MEMORYLESS POLICY IN THE DOUBLE-TAP ENVIRONMENT

If the agent follows the memoryless policy, its expected single-step return is[7]

$$E_\pi[r] = \Pr(s = s_1 \text{ and } a = a_1) + 2\Pr(s = s_2 \text{ and } a = a_2)$$
$$= p^2 + 2(1 - p)^2.$$

The full discounted reward is then given by

$$J(p) = E\left[\sum_{t=0}^{\infty} \gamma^t r_t\right]$$
$$= \sum_{t=0}^{\infty} \gamma^t E[r_t]$$
$$= \left(p^2 + 2(1 - p)^2\right) \sum_{t=0}^{\infty} \gamma^t$$
$$= \frac{p^2 + 2(1 - p)^2}{1 - \gamma}.$$

As is visible from the plot in Figure 2a, this function is convex and not concave, so gradient ascent may get stuck in local maxima. When the initialization $p_0$ is less than the minimum point $p = 2/3$, then gradient ascent converges to the global maxima $p = 0$. But if $p_0 > 2/3$, then it converges to the local maxima $p = 1$. So if $p_0$ is initialized uniformly at random on $[0, 1]$, gradient ascent converges to the suboptimal policy $\pi(a_1) = 1$ one third of the time.

---

[7]This isn't quite true for the first reward, where the value is simply $p$ or $2(1 - p)$, depending on whether we start in state $s_1$ or $s_2$. We ignore this for simplicity, since the effect on the training dynamics is negligible.

### A.1.2 EXPECTED RETURN OF A MOLICY WITH MEMORY IN THE DOUBLE-TAP ENVIRONMENT

We will now compute the expected return for a policy with one-step memory in the double-tap environment from Figure 1. This is the return plotted in Figure 2b.

First, note that the state $s^{(t)}$ at time $t$ is fully determined by the action $a^{(t-1)}$: if $a^{(t-1)} = a_1$, then $s^{(t)} = s_1$, and similary for $a_2$ and $s_2$. Thus a policy with memory $m_t = a^{(t-1)}$ is equivalent to a policy that can observe the state $s^{(t)}$.

We parametrize the agent via the 'state switch' probabilities $\theta \in \mathbb{R}^2$, such that

$$\pi_\theta(a_1 \mid s_2) = \theta_1$$
$$\pi_\theta(a_2 \mid s_1) = \theta_2.$$

Our goal is to compute the expected return

$$J(\theta) = \mathrm{E}\left[\sum_{t=0}^\infty \gamma^t r_t\right]$$
$$= \sum_{t=0}^\infty \gamma^t \mathrm{E}[r_t]$$
$$= \frac{1}{1-\gamma} \mathrm{E}[r_t],$$

where $r_t$ is the reward received in timestep $t$. Recall that $r_t = 1$ if the agent takes action $a_1$ from state $s_1$, $r_t = 2$ if the agent takes action $a_2$ from state $s_2$, and $r_t = 0$ otherwise. Thus

$$\mathrm{E}[r_t] = P(s_{(t)} = s_1)(1 - \theta_2) + 2P(s_{(t)} = s_2)(1 - \theta_1), \tag{4}$$

where $P(s_{(t)} = s)$ is the probability of being in state $s$ at time $t$. This distribution depends on the policy.

**Proposition A.2.** *As $t \to \infty$, the state distribution converges to*

$$P_\infty(s_i) = \lim_{t\to\infty} P(s_{(t)} = s_i) = \frac{\theta_i}{\theta_1 + \theta_2},$$

*for $i \in \{1, 2\}$.*

We substitute the stable state distribution given in Proposition A.2 into Equation 4:

$$E[r_t] = \frac{\theta_1(1 - \theta_2) + 2\theta_2(1 - \theta_1)}{\theta_1 + \theta_2}.$$

Thus the expected return (after acting in the environment for long enough that the state distribution has stabilized) is

$$J(\theta) = \frac{\theta_1(1 - \theta_2) + 2\theta_2(1 - \theta_1)}{(\theta_1 + \theta_2)(1 - \gamma)}. \tag{5}$$

This expression is plotted in Figure 2b.

*Proof of Proposition A.2.* A simple computation suffices to check that $P_\infty$ is the unique distribution that satisfies the fixed-point equation

$$P_\infty(s_1) = P_\infty(s_1)\pi_\theta(a_1 \mid s_1) + P_\infty(s_2)\pi_\theta(a_1 \mid s_2).$$

It remains to show that this fixed point iteration converges. To do this, consider the sequence $(p_t)_{t\in\mathbb{N}}$, where $p_t = P(s_t = s_1)$. Then the above fixed point iteration can be written as

$$p_{t+1} = p_t\pi(a_1 \mid s_1) + (1 - p_t)\pi(a_1 \mid s_2).$$

The corresponding transformation that maps $p_t$ to $p_{t+1}$ is

$$F\colon p \mapsto p\pi(a_1 \mid s_1) + (1 - p)\pi(a_1 \mid s_2).$$

It remains to show that $F$ is a contraction. Let $p, q \in [0, 1]$, then

$$
\begin{aligned}
|Fp - Fq| &= \left| (p - q)\pi(a_1 \mid s_1) + (q - p)\pi(a_1 \mid s_2) \right| \\
&= \left| (p - q)\left( \pi(a_1 \mid s_1) - \pi(a_1 \mid s_2) \right) \right| \\
&\leq |p - q| \cdot \left| \pi(a_1 \mid s_1) - \pi(a_1 \mid s_2) \right| \\
&\leq |p - q|.
\end{aligned}
$$

Thus $F$ is a contraction, and so $p_t$ must converge to the fixed point $p_\infty = \theta_2/(\theta_1 + \theta_2)$. $\qquad\square$

## A.2 BIT GUESSING

This environment has one hidden state $h \in \{0, 1\}$, observation space $\mathcal{O} = \{0, 1\}$ and action space $\mathcal{A} = \{0, 1\}^2$. The goal of the agent is to 'guess' the value of the hidden state, given an observation that is either equal to the hidden state ($o = h$) or bit-flipped ($o = 1 - h$).

In each step, the hidden state is sampled uniformly and independently of any past actions. The agent takes an action $a = (a_{\text{guess}}, a_{\text{flip}}) \in \{0, 1\}^2$. Reward is determined by

$$
r(a, h) = \begin{cases} 1 \text{ if } a_{\text{guess}} = h \\ 0 \text{ else.} \end{cases}
$$

The value of $a_{\text{flip}}$ determines the observation at the next timestep: if $a_{\text{flip}} = 1$, then the next observation is flipped; otherwise it is equal to the hidden state.

An RL algorithm training in this environment has two equilibria: one where the policy never takes action $a_{\text{flip}}$ and sets its guess equal to the observation, and one where the policy consistently takes $a_{\text{flip}}$ and always guesses $a_{\text{guess}}^{(t)} = 1 - o^{(t)}$.

## A.3 BATTLE OF THE SEXES

Any two-player game can be turned into a 'single-player' RL environment by having the RL agent play against a fixed policy. We choose an opponent policy that is able to see the history (in contrast to the agent), and always chooses the action that would have done best against previous actions within an episode. More details in Appendix B.

### EXPECTED RETURN IN BATTLE OF THE SEXES

Let $\mathcal{A}$ be the (finite) action space, and for $a \in \mathcal{A}$ let $r_a$ and $r_a^{\text{opp}}$ be the rewards for agent and "opponent" respectively, given that both play action $a$.

The expected return of a memoryless policy $\pi$ in timestep $t$ is equal to

$$
R_t(\pi) = \sum_{a \in \mathcal{A}} \pi(a) r_a \Pr(a_t^{\text{opp}} = a), \tag{6}
$$

where $\Pr(a_t^{\text{opp}} = a)$ is the probability that the opponent will play $a$ in timestep $t$. The opponent play depends on past actions: the opponent always plays the action

$$
a_t^{\text{opp}} = \text{argmax}_{a \in \mathcal{A}} \sum_{t' \leq t} r_a^{\text{opp}} \cdot \mathbb{I}(a_{t'} = a)
$$

that plays best against the agent's past actions within an episode. Setting the 'count' variables $c(a) = \sum_{t' \leq t} \cdot \mathbb{I}(a_{t'} = a)$, we can write this as

$$
a_t^{\text{opp}} = \text{argmax}_{a \in \mathcal{A}} \, c(a) \cdot r^{\text{opp}}(a).
$$

The opponent (i.e. environment) play $a_t^{\text{opp}}$ is random, since it depends on the (random) episode history and in particular on $c(a)$.

We will work out the expected return for the special case $\mathcal{A} = \{a_1, a_2\}$, the standard 'battle of the sexes' game with two options. In this case, writing $c = c(a_1)$ and noting that $c(a_2) = t - c(a_1)$, and that $c$ is a binomial random variable,

$$
\begin{aligned}
\Pr(a_t^{\mathrm{opp}} = a_1) &= \Pr(c(a_1) \cdot r_1^{\mathrm{opp}} > c(a_2) \cdot r_2^{\mathrm{opp}}) \\
&= \Pr(c r_1^{\mathrm{opp}} - (t - c) r_2^{\mathrm{opp}} > 0) \\
&= \Pr(c > l) \\
&= \sum_{k=l+1}^{t} \binom{t}{k} \pi(a_1)^k (1 - \pi(a_1))^{t-k}
\end{aligned}
$$

where $l = \lfloor t r_2^{\mathrm{opp}} / (r_1^{\mathrm{opp}} + r_2^{\mathrm{opp}}) \rfloor$.

## B  CONNECTIONS BETWEEN NASH EQUILIBRIA AND RL TRAINING EQUILIBRIA

Any two-player game can be turned into a 'single-player' RL environment by having the RL agent play against a fixed 'opponent' policy. The opponent policy is able to see the agent's past actions and always chooses the action that would have done best against the agent's previous actions within an episode.

In this section we formalize this idea and show that Nash equilibria correspond to stationary points of the expected return (and thus are equilibria of the policy gradient algorithm). In particular, any game with multiple Nash equilibria (e.g. 'Battle of the Sexes') will also have multiple training equilibria when turned into an RL environment in this way.

**Definition B.1.** A *two-player game* is a tuple $(\mathbf{S}, \mathbf{u})$, where $\mathbf{S} = (S_1, S_2)$ is a tuple of pure strategy sets $S_i = \{1, \ldots, k_i\}$ and $\mathbf{u} = (u_1, u_2)$ is a tuple of payoff functions $u_i \colon S_1 \times S_2 \to \mathbb{R}$.

Thus the payoff of player $i$ is given by $u_i(s_1, s_2)$, where $s_i \in S_i$ are the strategies of players 1 and 2 respectively. We now construct a POMDP by putting an RL agent in the role of player 1 and letting the environment control player 2.

**Definition B.2.** Let $G = (\mathbf{S}, \mathbf{u})$ be a two-player game. Fix the episode length $T > 0$. We construct a POMDP from the game $G$ by defining $E = (\mathcal{S}, \mathcal{A}, \mathcal{O}, T, O, R, \gamma, \eta_0)$ in the following way; set $\mathcal{A} = S_1$, and let the state space $\mathcal{S}$ be the set of all possible action histories of maximum length $T$, that is

$$
\mathcal{S} = \bigcup_{t=1}^{T} \mathcal{A}^t,
$$

where

$$
\mathcal{A}^t = \underbrace{\mathcal{A} \times \cdots \times \mathcal{A}}_{t \text{ copies of } \mathcal{A}}.
$$

We choose the observation space $\mathcal{O} = \emptyset$ to be empty, thus the distribution of observations $O$ is trivial. The transition function $T$ amounts to appending the current action to the history: for any history $h = (a_1, \ldots, a_{t-1})$ and action $a_t$ the transition kernel is deterministic and given by

$$
T(h' \mid h, a) = \begin{cases} 1 \text{ if } h' = h \cup (a) \\ 0 \text{ otherwise.} \end{cases} \tag{7}
$$

To define the reward function $R$, we need to set the strategy of the opponent, which we choose to be the average best-response over the episode history: for any timestep $t > 1$ after the first,

$$
a_t^{\mathrm{opp}} = \mathrm{argmax}_{a \in S_2} \sum_{t' < t} u_2(a_t, a).
$$

In the first timestep, $a_1^{\mathrm{opp}}$ is sampled uniformly from $S_2$. For any history $h \in \mathcal{S}$ and action $a \in \mathcal{A}$ the reward is equal to the payoff of player 1 in the current timestep, that is $R(h, a) = u_1(a, a^{\mathrm{opp}})$, where $a^{\mathrm{opp}}$ is the opponent action chosen as in Equation 7 according to the history $h$.

For convenience, we repeat the definition of a Nash equilibrium:

**Definition B.3** (Nash Equilibrium). Let $G = (\mathbf{S}, \mathbf{u})$ be a two-player game. A *Nash equilibrium* is a pair of strategies $(s_1, s_2) \in S_1 \times S_2$ such that for all $s_1'$, $s_2'$ in $S_1$ resp. $S_2$,

$$u_1(s_1', s_2) \leq u_1(s_1, s_2)$$
$$u_2(s_1, s_2') \leq u_2(s_1, s_2).$$

In the Battle of the Sexes environment, every Nash equilibrium in the original two-player game corresponds to an RL training equilibrium in the corresponding POMDP. We now prove that this is true for all two-player games.

**Theorem B.1.** *Let $G = (\mathbf{S}, \mathbf{u})$ be a two-player game and $E$ its associated POMDP. Then the following are equivalent:*

*(i) The policy $\pi$ is a policy-gradient equilibrium in $E$.*

*(ii) The strategy set $(s^\pi, s^{\pi,opp})$ is a Nash equilibrium in $G$.*

*Proof.* For any $s \in S_2$, define the environment $E_s$ by taking a copy of $E$ and replacing the best-response opponent strategy with the strategy $s$. Since $s$ is chosen independently of past actions, $E_s$ is an MDP and standard convergence guarantees hold: policy gradient always learns an optimal policy. Conversely, any optimal strategy is also a policy-gradient equilibrium point.

$(i) \implies (ii)$: First, consider the environment $E_{s_\pi^{opp}}$. Since $\pi$ is in equilibrium, it must be an optimal policy for $E_{s_\pi^{opp}}$. Thus for all $s' \in S_1$,

$$u_1(s', s_\pi^{opp}) \leq u_1(s^\pi, s_\pi^{opp}).$$

Second, by definiton of the best-response strategy $s_\pi^{opp}$ there is no strategy that can do better; thus for all $s' \in S_2$

$$u_2(s^\pi, s') \leq u_2(s^\pi, s_\pi^{opp}).$$

We conclude that $(s^\pi, s_\pi^{opp})$ is a Nash equilibrium.

$(ii) \implies (i)$: Assume $(s^\pi, s^{\pi,opp})$ is a Nash equilibrium. This implies $\pi$ is an optimal policy for $E_{s_\pi^{opp}}$; in particular it is also an equilibrium point. Since $s^{opp}$ is the optimal response strategy, $\pi$ is also a policy-gradient equilibrium point in $E$. $\square$

## C  WHEN DO Q-LEARNING AND POLICY GRADIENT METHODS CONVERGE IN POMDPS?

For convenience we repeat the usual conditions for convergence (apart from full observability). In the following we will always assume they are met.

1. Full exploration (every state-action pair is visited infinitely often).

2. The rewards are bounded: there exists some $R_{\max}$ such that for all state-action pairs $(s, a) \in \mathcal{S} \times \mathcal{A}$ we have $R(s, a) \leq R_{\max}$.

3. Flexible enough representation.

   - For policy gradient: the policy space is closed under policy improvement (Bhandari and Russo 2019).
   - For Q-learning: the representation of the Q-function is tabular (Watkins and Dayan 1992).

4. The learning rates satisfy the standard Robbins-Monro conditions: for all state-action pairs $(s, a)$, the learning rate $(\alpha_n)_{n \geq 0}$ satisfies $\sum_{i=0}^\infty \alpha_{n(i,s,a)} = \infty$ and $\sum_{i=0}^\infty \alpha_{n(i,s,a)}^2 < \infty$. Here, $n(i, s, a)$ is the index corresponding to the $i$th time that the agent visits state $s$ and takes action $a$.

## C.1 THE CONDITIONAL STATE DISTRIBUTION IS UNAFFECTED BY THE POLICY

**Proposition C.1.** *Let $E = (\mathcal{S}, \mathcal{A}, \mathcal{O}, T, O, R, \gamma, \eta_0)$ be a finite POMDP. For any policy $\pi$, let $\eta^\pi$ be the distribution on states induced by the policy, and let*

$$\eta^\pi(s \mid o) = \frac{O(o \mid s)\eta^\pi(s)}{\sum_{s' \in \mathcal{S}} O(o \mid s')\eta^\pi(s')}$$

*be the conditional distribution of state $s \in \mathcal{S}$ given an observation $o \in \mathcal{O}$.*

*Assume that the conditional distribution is independent of the policy in the sense that there exists some $\eta$ such that for all state-observation pairs $\eta(s \mid o) = \eta^\pi(s \mid o)$. Further assume that all the conditions listed in the beginning of Appendix C hold. Then Q-learning and policy gradient are guaranteed to converge to an optimal policy.*

*Proof.* The main idea is that we can construct an MDP $E'$ that is equivalent to the POMDP $E$ in the sense that it generates the same trajectories. Then the standard convergence guarantees apply.

Indeed, consider the MDP $E' = (\mathcal{S}', \mathcal{A}, T', R', \gamma, \eta_0)$, where $\mathcal{S}' = \mathcal{O}$. The transition function $T'$ is defined such that for $o', o \in \mathcal{O}$ and $a \in \mathcal{A}$,

$$T'(o' \mid o, a) = \sum_{s', s \in \mathcal{S}} O(o' \mid s')T(s' \mid s, a)\eta(s \mid o)$$

and the stochastic reward is given by

$$R'(o, a) = R(s, a)$$

where $s \sim \eta(s \mid o)$. By construction, the dynamics of $E$ and $E'$ are equivalent: indeed, all we've done is replace the transition function $T$ with a new kernel that marginalizes over states, and made the reward function stochastic.

Standard results for convergence in MDPs (Bhandari and Russo 2019; Watkins and Dayan 1992) now suffice to show that Q-learning and policy gradient learn an optimal policy in $E'$ and thus in $E$. $\quad\square$

## C.2 SUFFICIENT POLICY MEMORY

The core problem in POMDPs is that the distribution $\eta^\pi(s \mid o)$ of the unobserved state given an observation depends on the policy $\pi$. We should be able to fix this by conditioning on past observations and actions as well: that is, we equip the policy with memory.

In this section we will show that if a policy has unlimited memory (that is, it can condition on the entire history) then RL algorithms always learn an optimal policy (Proposition C.2) in all POMDPs. In addition, if the observed environment is $k$-Markov then the memory only needs to be able to store the past $k$ timesteps.

**Proposition C.2.** *Let $(\mathcal{S}, \mathcal{A}, \mathcal{O}, T, O, R, \gamma, \eta_0)$ be a POMDP. Consider a parametrized policy $\pi_\theta$ that has memory, that is it can condition on all past states and observations: the action $a_t$ at time $t$ is sampled from $\pi_\theta(\cdot \mid o_0, a_0, \ldots, o_{t-1}, a_{t-1}, o_t)$. If this policy is trained using tabular Q-learning or VPG it benefits from the standard convergence guarantees (Bhandari and Russo 2019; Watkins and Dayan 1992) as in MDPs.*

The main idea here is to show that we can construct an MDP whose states consist of the entire POMDP history, and then show that this is equivalent to an agent with sufficient memory. Let $E = (\mathcal{S}, \mathcal{A}, \mathcal{O}, T, O, R, \gamma, \eta_0)$ be a POMDP as in Proposition C.2

We will now construct an MDP $E' = (\mathcal{S}', \mathcal{A}', T', R')$ such that training a memoryless policy in $E'$ is equivalent to the training setup in Proposition C.2.

Set $\mathcal{S}'$ to be the set of all states of the form $s' = (o_0, a_0, \ldots, o_{t-1}, a_t, o_t)$ for $t \leq T$ where $T$ is the maximum number of timesteps within an episode. Let $\mathcal{A}' = \mathcal{A}$. Let $T(s', a')$ be the distribution over states $s''$ such that $s'' = s' \cup (a', o)$, where $\cup$ is the 'extend' operation and $o$ is the observation from $E$ distributed conditional on the history $s' \cup (a')$. Finally, set the reward $R'(s', a')$ to be a random variable $R'(s', a') = R(s, a')$, where the POMDP state $s$ is distributed conditional on the history $s' \cup (a')$.

Now $E'$ is a well-defined MDP with stochastic rewards. Well-known results (Watkins and Dayan 1992; Bhandari and Russo 2019) now imply the convergence guarantees we wanted.

**Proposition C.3.** *Let $E = (\mathcal{S}, \mathcal{A}, \mathcal{O}, T, O, R, \gamma, \eta_0)$ be a POMDP. Assume that the environment state is $k$-Markov in the following sense: at timestep $t$, conditional on the length-$k$ history $h_{t-k}^t = (o_{t-k}, a_{t-k}, \ldots, a_{t-1}, o_t)$, the state $s_t$ is independent of the full history: $(s_t \perp h_0^t) \mid h_{t-k}^t$.*

*Consider a parametrized policy $\pi_\theta$ that has memory of length $k$: action $a_t$ at time $t$ is sampled from $\pi_\theta(\cdot \mid h_{t-k}^t)$. If this policy is trained using tabular Q-learning or VPG it benefits from the standard convergence guarantees (Bhandari and Russo 2019; Watkins and Dayan 1992) as in MDPs.*

*Proof.* □

Just as previously, we construct an MDP $E' = (\mathcal{S}', \mathcal{A}', T', R')$ such that training a memoryless policy in $E'$ is equivalent to the training setup in Proposition C.3.

Set $\mathcal{S}' = \{h_{t-k}^t \mid t \leq T\}$ to be the set of all $k$-histories. Here $T$ is the maximum number of timesteps within an episode.

Let $\mathcal{A}' = \mathcal{A}$. For all $t \leq T$, let $T(h_{t-k}^t, a)$ be the distribution over $k$-histories $h_{t-k+1}^{t+1}$ that results from taking action $a$ at step $t$.

Let $h, a$ be a $k$-history and an action, and $o$ a random observation drawn from the distribution

$$p(o \mid h, a) = \sum_{s', s'' \in \mathcal{S}} O(o \mid s'') T(s'' \mid s', a) p(s' \mid h).$$

(Because the enviornment is $k$-Markov, the conditional distribution $\eta(s \mid h)$ does not depend on actions or observations more than $k$ timesteps in the past.)

Then we can set the new transition dynamics $T'(h, a)$ to be the distribution over $k$-histories $h'$ such that $h'$ is the result of appending $a$ and $o$ to the end of the history $h$ and removing the first timestep recorded in $h$.

Finally, set the reward $R'(h, a)$ to be a random variable $R'(h, a) = R(s, a)$, where the POMDP state $s$ is distributed conditional on the history $h$.

Standard results for convergence in MDPs (Bhandari and Russo 2019; Watkins and Dayan 1992) now suffice to show that Q-learning and policy gradient learn an optimal policy in $E'$ and thus in $E$.

