# OpenReview forum: "Training Equilibria in Reinforcement Learning"
_ICLR.cc/2023/Conference — Submitted to ICLR 2023_

### Official Review · Reviewer_CVMT · 2022-10-22

**Confidence:** 3
**Correctness:** 4
**Technical Novelty And Significance:** 3
**Empirical Novelty And Significance:** 2
**Recommendation:** 5

**Clarity, Quality, Novelty And Reproducibility:**

Clarity:
The order of concepts in the paper could be improved. In particular, I would recommend a background section so that the relevant game theoretic concepts are introduced before they are referred to.
Otherwise, clarity is a strength of the paper.

Quality:
The paper perhaps pushes a little hard on the potential impact of its results.
The main claims are clearly justified.

Novelty:
As far as I know, the authors observations are novel.

Reproducibility:
The paper provides enough detail. I don't think that a code release is necessary for this kind of paper.

**Strength And Weaknesses:**

Strengths:
1. The narrative, mathematics and experiments of the paper are very clear and logical.
2. Issues relating to training RL in POMDPs are salient to many. The paper gives logical justification for some of these issues.

Weaknesses:
1. There is not a lot of depth to the observations—the connection between POMDPs and coordination games is fairly surface-level. There is a lot more to POMDPs than those that can be solved optimally with a few states of memory. For example, classic robot control-style with sensor and location uncertainty. I can imagine two ways to improve the paper:
- Expand the experiments section to more complex environments that still have the interesting properties and measure the impact there
- Increase the strength of the theoretical connection—a potential avenue for this is to think about connections to evolutionary game theory

**Summary Of The Paper:**

The authors draw an analogy between memoryless policies in POMDPs and coordination games. In a POMDP, a memoryless policy can be viewed as playing a game with itself in the past---because it can't remember its past action, it is as if it is playing a one-shot game. The authors show that there is a class of POMDPs where this connection is direct---you can convert between the game and the POMDP and the equilibria are the same in each (the authors define a notion of training equilibria in these memoryless POMDPs).

The authors claim that this observation provides a new justification for why noise in parameter space is valuable (something that had previously been found empirically)—it can remove local maxima.

In experiments, the authors explore the effect of parameter noise and the effect of adding additional memory in simple settings. Memory is helpful, but the algorithms do not use it efficient, and too much memory degrades performance.

**Summary Of The Review:**

The paper's main contributions tell us something interesting about current practices in RL for POMDPs, but they are somewhat surface-level. The paper is presented in a clear and accessible manner.

---

### Official Review · Reviewer_A2qp · 2022-10-23

**Confidence:** 3
**Clarity, Quality, Novelty And Reproducibility:** The code to reproduce the numerical r…
**Correctness:** 4
**Technical Novelty And Significance:** 4
**Empirical Novelty And Significance:** 3
**Recommendation:** 6

**Strength And Weaknesses:**

Strength:
The authors show theoretically that multiple equilibria can only emerge when the distribution of unobserved states depends on past actions. A sufficient amount of memory resolves the coordination game and provably implies a unique optimal training equilibrium.

Weaknesses:
More empirical results are expected to show the practical application of the theory.


**Summary Of The Paper:**

In partially observable environments, reinforcement learning algorithms such as policy gradient and Q-learning can converge to equilibria that are strictly suboptimal. The authors point the core problem is that in partially observed environments, an agent’s past actions induce a distribution on hidden states. This problem can be solved by unbounded memory, and may be alleviated by bounded memory.

**Summary Of The Review:**

The problem is interesting, and the theoretical analysis is insightful.

---

### Official Review · Reviewer_C66v · 2022-10-24

**Confidence:** 4
**Correctness:** 3
**Technical Novelty And Significance:** 2
**Empirical Novelty And Significance:** 2
**Recommendation:** 3

**Clarity, Quality, Novelty And Reproducibility:**

Clarity is lacking in places, but overall not bad. Certain details are missing.

Novelty is not very high.

Reproducibility is not very good due to missing details like hyperparameters and some details of the environments used.

See above for more details.

**Strength And Weaknesses:**

Strengths
=========
The analogy between Nash equilibria and learning equilibria in POMDPs is somewhat interesting and makes intuitive sense.

The point that memory is useful in POMDP learning even if the optimal policy does not require memory is interesting and valid.

The use of parameter noise to escape suboptimal training equilibria is interesting, if not surprising.

Weaknesses
==========
The result that algorithms like Q-learning and policy gradient may have multiple training equilibria is not entirely novel. It is essentially implied by the existence of multiple greedy partitions highlighted in section 6.4 by Bertseka (1996), for example. This prior work is not cited, nor is more recent work on the topic by the same author (Bertsekas 2011). For policy gradient, this result is explicitly given in the cited work of Bhandari and Russo (2019). Though as the authors mention it is in the context of a constrained policy class rather than partial observability, the equivalence between these two cases is well known. Young and Sutton (2020) also highlight the possibility of multiple training equilibria explicitly for Q-learning and policy gradient, which is not cited, but given it seems to only appear on arxiv this omission is forgivable.

Proposition 3.1 states that when the observation conditional state probabilities are policy independent, Q learning and policy gradient are guaranteed to converge to an optimal policy in a POMDP. I think what is meant is an optimal memoryless policy, but this is very confusing because no clear distinction is made and elsewhere adding memory is shown to allow convergence to the "optimal policy".

The result that including sufficient memory allows convergence to the optimal policy in a POMDP under appropriate circumstances is well known and hence I don't feel it is a significant contribution. This amounts to saying that with sufficient memory a POMDP becomes an MDP and thus normal guarantees for MDPs apply. This is true generally if one conditions on the full history, and true with a small "length-k history" essentially by definition if the environment is k-Markov.

The result in theorem 3.2, that Nash equilibria of two-player games correspond to training equilibria of a related RL problem is unclear to me and I believe incorrect as stated in the appendix. The MDP is defined by running some arbitrary fixed number of iterations of the two-player game where one opponent (i.e. the environment player) is effectively fixed to play the best response to the actions taken so far by the other (i.e. the "agent" player). It seems to be assumed that the environment is then the best response to the policy of the agent player, but this is clearly not true in general. For example, it is obviously untrue if the episode length is 1 as the environment's first action is assumed to be random. It may be true in the limit of infinite episode length, but the present argument is not sufficient to show this. Aside from that, I'm not entirely clear what the point of this theorem is as it doesn't seem to help to explain or motivate anything else in the paper aside from the hand-wavy statement "In practice, the analogy seems to extend to contexts outside of self-play or those covered in Theorem 3.2". I think more work needs to be done to motivate this connection.

Overall I found the connection between Nash equilibrium and POMDP learning to be pretty vague in terms of how it is presented in the current paper. In particular, prior to introducing the theorem, it's loosely implied that one can understand POMDPs generally in terms of Nash equilibria. Instead what is supposedly shown is that one can construct a POMDP from a given 2-player game in a particular way and then show a correspondence between Nash equilibria of the game and training equilibria of the POMDP which is quite different and in my mind less interesting.

Nothing is said about hyperparameters or how they were chosen for the empirical results, which may be important for interpreting them since I would expect, for example, the performance of PPO with larger numbers of memory bits to depend on these details. Similarly, some details of the environments are missing, including the maximum length of the episodes and how exactly the bit-flipping actions are added to the action space where applicable.

Minor Comments, Corrections and Questions
=========================================
* In appendix A it says "(TODO ref figure)" which presumably should point to a figure instead.

* In section 3 "continuous action selection" is kind of vague and could be clarified.

* In 3.1 A(\pi_{\theta_k})=\pi_{\theta_{k+1}} should probably operate on the parameters rather than the policy if it is meant to be general enough to include Q-learning. For example in Q-learning one may update the parameters in a way that changes the value, but leaves the (epsilon-)greedy policy the same.

* In the abstract, it says "Prior work blames insufficient exploration" but I can't see that this is ever mentioned again and no citation to the specific prior work that does so is given, so this could be taken as an unsubstantiated claim.

* In 3.3 it says that a Nash equilibrium is a solution to a "non-cooperative game", but later it says "Nash equilibrium of a cooperative game" which is a weird inconsistency.

* Missing reference (??) in 3.3.

* Missing period before "Unfortunately" in 4.2.

* It would be good to show a line corresponding to optimal performance in Figure 4.

* In Appendix B T is overloaded to mean both the transition kernel and the episode length.

* What episode length is used in the experiments in Section 5.2?

* In the sequential bandit problem is the bit flip added as an extra action, or is the action the composition of the original actions and optionally flipping each bit?

References
==========
Young, Kenny, and Richard S. Sutton. "Understanding the pathologies of approximate policy evaluation when combined with greedification in reinforcement learning." arXiv preprint arXiv:2010.15268 (2020).

Bertsekas, Dimitri, and John N. Tsitsiklis. Neuro-dynamic programming. Athena Scientific (1996).

Bertsekas, Dimitri P. "Approximate policy iteration: A survey and some new methods." Journal of Control Theory and Applications 9.3 (2011): 310-335.

Bhandari, Jalaj, and Daniel Russo. "Global optimality guarantees for policy gradient methods." arXiv preprint arXiv:1906.01786 (2019).

**Summary Of The Paper:**

This paper points out that under partial observability algorithms like policy gradient and Q-learning may have multiple training equilibriums, though this observation is, in itself, not entirely novel. On top of this, they present a number of other results and proposals including:

* Showing multiple training equilibria can only arise when the observation conditional state distribution depends on the policy and thus the issue can be understood as a consequence of this dependence.

* Suggesting a correspondence between Nash equilibria and training equilibria.

* Showing that by adding sufficient memory to the state implies convergence to the unique optima.

Experiments are also conducted in simple toy environments which demonstrate how adding memory can improve performance in partially observable domains even if there is an optimal memoryless policy. Further experiments demonstrate how adding parameter noise can allow an agent to converge reliably to better training equilibria when multiple are present, at least in some simple environments.

**Summary Of The Review:**

This paper might have the beginning of some interesting directions, but I feel it is far from ready in its current form. Most of the theory is either only vaguely related to the stated point, or already widely known. The empirical results are similarly quite simple and not very robust in their conclusions.

---

### Official Review · Reviewer_QaRE · 2022-10-28

**Confidence:** 3
**Correctness:** 3
**Technical Novelty And Significance:** 2
**Empirical Novelty And Significance:** 2
**Recommendation:** 3

**Clarity, Quality, Novelty And Reproducibility:**

-) The writing could be improved in my opinion. The different sections do not naturally add up to a single coherent message. Further, the assumptions that were made (in proposition 3.1.) are not properly motivated. The example the authors focus on in figure 1 is quite basic and do not represent a common class fo interesting POMDPs. Are there any interesting POMDPs that satisfy thes assumption?

-) I find the novelty of the paper restricted at this point. The theoretical results are basic (exploration is not tackled, asymptotic analysis, additional hard assumptions) and the bottom line is quite expected (that memory improves the convergence ability). On the other hand, the empirical result are also quite basic.


**Strength And Weaknesses:**

Strength.

-) The POMDP setting is of importance. To make RL practically useful it is much needed to improve our understanding and tools. This work established the convergence of two of the most popular RL algorithms for POMDDs under assumptions.

Weaknesses.

-) I find the assumptions made in proposition 3.1. quite harsh. Further, the fact there's no finite time analysis nor explicit treatment of the exploration problem weakens the theoretical results in my opinion.

-) The study of this work is not coherent in my opinion. The theoretical results, in my opinion, are not very much related to one another. It seems there's no concrete message, but rather couple of messages that do not necessarily add together to a coherent story/

-) The experimental results are quite basic. The experiments do not show of new capability, or achieve state of the art performance. Hence, the experimental part does not add much more over the theoretical part.

**Summary Of The Paper:**

This paper studies several aspects of RL with non-markovian observations under the assumption there is no exploration issue (assumption 3.1) while focusing on asymptotic guarantees. The authors establish several results which are its key contributions:
1. Under several (quite harsh) assumptions, Q learning and policy gradient converge to an optimal policy (proposition 3.1.).
2. A correspondence between POMDPs and Nash equilibrium (theorem 3.2.).
3. Memory based policies improve the asymptotic convergence (proposition 4.1).
Further, the authors conduct some simple experiments to exemplify these observations.



**Summary Of The Review:**

Due to the above reasons I believe this work is not yet ready for publications. Specifically:
1) The theoretical and experimental results are basic.
2) I currently do not find the different results related to one another.
3) The main motivation is a toy example. The authors should show (in my opinion) it captures additional interesting problems.

I also attach couple of questions that bothered me while reading thus workL
Page 2. Nash equilibria in POMDPs
Proposition 3.1:
1) Shouldn't \eta have time index? that is, the probability the state at time t is s conditioning on the observation at time t is o?
2) The assumption here is quite harsh in my opinion and restrict the result to a very specific structure (since \eta is not a function of pi for all \pi). Which POMDPs satisfy this assumption? I can see that if the latent state is decodable this assumption is satisfied, but then the environment is not partially observed.
3) The policy class includes the optimal policy? is it the set of all policies?

---

### Decision · Program_Chairs · 2023-01-20

**Decision:**

Reject

**Justification For Why Not Higher Score:**

The results don't seem to be significant enough to meet the bar for ICLR.

**Justification For Why Not Lower Score:**

n/a

**Metareview: Summary, Strengths And Weaknesses:**

(a) Summary:  The paper draws an analogy between multi-agent reinforcement learning in partially observable MDPs and coordination games.  Standard RL algorithms can converge to suboptimal equilibria, but adding memory can help.

(b) Strengths: All reviewers agreed that the paper considers an important problem, as POMDPs are an important setting.  Multiple reviewers found the use of parameter noise to avoid or escape suboptimal equilibria interesting.

(c) Weaknesses: Several reviewers found the experimental results unconvincing.  Most reviewers had serious concerns about the novelty and significance of the main theoretical results.  There were widespread concerns about clarity.


**Summary Of Ac-Reviewer Meeting:**

n/a